# Effects of Mediterranean Diet Combined with CrossFit Training on Trained Adults’ Performance and Body Composition

**DOI:** 10.3390/jpm12081238

**Published:** 2022-07-28

**Authors:** Salvatore Ficarra, Domenico Di Raimondo, Giovanni Angelo Navarra, Mohammad Izadi, Alessandra Amato, Francesco Paolo Macaluso, Patrizia Proia, Gaia Musiari, Carola Buscemi, Anna Maria Barile, Cristiana Randazzo, Antonino Tuttolomondo, Silvio Buscemi, Marianna Bellafiore

**Affiliations:** 1Sport and Exercise Sciences Research Unit, Department of Psychology, Educational Science and Human Movement, University of Palermo, Via Giovanni Pascoli 6, 90144 Palermo, Italy; salvatore.ficarra03@unipa.it (S.F.); giovanniangelo.navarra@unipa.it (G.A.N.); 2016if@gmail.com (M.I.); alessandra.amato02@unipa.it (A.A.); francesco.macaluso91@libero.it (F.P.M.); patrizia.proia@unipa.it (P.P.); marianna.bellafiore@unipa.it (M.B.); 2Department of Health Promotion, Mother and Child Care, Internal Medicine and Medical Specialties (ProMISE) “G. D’Alessandro”, University of Palermo, Piazza delle Cliniche, 90127 Palermo, Italy; gaiamusiari@gmail.com (G.M.); carola.buscemi@gmail.com (C.B.); annamaria.barile@unipa.it (A.M.B.); cristiana.randazzo@unipa.it (C.R.); bruno.tuttolomondo@unipa.it (A.T.); silvio.buscemi@unipa.it (S.B.)

**Keywords:** Mediterranean diet, CrossFit, fitness, body composition

## Abstract

CrossFit is a high-intensity training discipline increasingly practiced in recent years. Specific nutritional approaches are usually recommended to maximize performance and improve body composition in high-intensity training regimens; notwithstanding, to date there are no targeted nutritional recommendations for CrossFit athletes. The Mediterranean Diet (MD) is a diet approach with a well-designed proportion of macronutrients, using only available/seasonal food of the Mediterranean area, whose health benefits are well demonstrated. No studies have evaluated this dietary strategy among CrossFit athletes and practitioners; for this reason, we tested the effects of 8 weeks of MD on CrossFit athletes’ performance and body composition. Participants were assigned to two groups: a diet group (DG) in which participants performed CrossFit training plus MD, and a control group (CG) in which participants partook in the CrossFit training, continuing their habitual diet. Participants were tested before and after the 8 weeks of intervention. At the end of the study, no significant difference was noted in participants’ body composition, whereas improvements in anaerobic power, explosive strength of the lower limbs, and CrossFit-specific performance were observed only in the DG. Our results suggest that adopting a MD in CrossFit athletes/practitioners could be a useful strategy to improve specific strength, endurance, and anaerobic capacity while maintaining overall body composition.

## 1. Introduction

CrossFit is a well-known, widely spread, new type of exercise training consisting of high-intensity and endurance activities, while presenting functional exercises as a major component. Thanks to CrossFit characteristics, athletes can simultaneously improve strength and aerobic performances and decrease body fat efficiently [1,2]. This type of training, usually composed of high-intensity exercise involving upper and lower body exercise with a short rest, especially when associated with uncontrolled progressive overload, may result in fatigue exacerbated by a non-suitable recovery that could subsequently lead to a higher risk of musculoskeletal injuries [2]. The athletes’ performance can be influenced by diet habits; several studies [3,4,5] have demonstrated that nutrients and supplements may have an important role in managing fatigue decrease and promoting recovery in athletes, which can be beneficial for the next training session. Given this, specific diet programs and targeted nutritional approaches should be considered for CrossFit athletes, especially when programmed competitions are taken under consideration. Nevertheless, only a few studies of low quality have assessed this important issue according to a recent review by Dos Santos Quaresma et al. [6], and the authors conclude there is a substantial lack of scientific evidence.

The traditional Mediterranean diet (MD) includes a high intake of cereals, vegetables, fruits, nuts, and olive oil; moderate consumption of dairy—mainly cheese—fish, and poultry; and low consumption of red and processed meats [7]. Higher adherence to the MD has been associated with some cardiometabolic benefits, including a lower incidence of metabolic syndrome and cardiovascular and cerebrovascular diseases [8,9,10,11]. This is mainly due to different mechanisms, such as lipid reduction, balancing oxidative stress and inflammation, regulation of growth hormones (involved in carcinogenesis), and many others [12]. CrossFit athletes need to consume no lesser than 5 to 8 g/kg (of body weight) of carbohydrates per day in order to restore liver and muscle glycogen due to the moderate to high intensity sustained during training [13]. From MD, athletes can receive an adequate intake of carbohydrates, which are appropriate to promote optimal recovery and improve CrossFit athletes’ performance, alongside bioactive compounds [14], which can lead to health promotion among athletes. Despite these potential benefits, to our knowledge, no study to-date has investigated whether a healthy dietary approach based on MD can lead to changes in athletic performance and body composition among CrossFit athletes or practitioners.

The aim of the current study was to compare the effects of an individualized dietary approach based on the Mediterranean diet, when added to habitual training, on the body composition and physical performance of CrossFit athletes.

## 2. Materials and Methods

A longitudinal experimental study was adopted to investigate the effects of an individualized MD-based food plan for CrossFit athletes on training-related variables such as explosive strength, muscular endurance, anaerobic power/capacity, and body composition.

### 2.1. Participants

CrossFit athletes/practitioners between 20 and 50 years with at least 1 year of regular CrossFit training experience with a minimum of 3 training sessions per week were included in the study. The inclusion criteria were healthy condition and not participating in drug treatment for any acute or chronic pathology. Thirty participants (15 men and 15 women), all volunteers from a class of CrossFit athletes/practitioners, were recruited for this study. All of them, at the time of enrollment, were following an uncontrolled diet without any supplementation. The participants were divided into 2 groups, the diet group (DG: 39.11 ± 8.9 years; 70.73 ± 10.4 kg) and the control group (CG: 35.58 ± 8.4 years; 72.9 ± 13.8 kg). The inclusion of enrolled subjects in the two groups was decided according to their individual availability to change their dietary habits. Notwithstanding, two homogeneous groups were obtained (no statistical differences were presented at baseline on anthropometric values and major variables analyzed). Only 22 (9 women, 13 men) completed the entire study protocol and were included in the final analysis: 6 male and 4 female participants in the DG, 7 male and 5 female participants in the CG. Written informed consent was obtained from all participants prior to the start of the study. All procedures were conducted in accordance with the ethical standards of the 1964 Declaration of Helsinki. The ethics committee of the University of Palermo approved the project (Verbale n. 2/2021). The study was conducted from May to August 2021 (8 weeks).

### 2.2. Intervention

The DG group followed a MD-based food plan [12], which was accurately individualized during the 8 weeks. Individual food plans were drafted by an experienced dietician who supervised the participants during the entire duration of the study. Individualization was achieved by calculating and adapting both energy intake (based on predicted energy expenditure) and macronutrient needs for each subject. The dietician adapted the diet prescribed to each subject according to individual requests. The participants in the CG did not follow any specific food plan and were asked to maintain their nutritional habits throughout the entire duration of the study.

The total daily energy expenditure of the participants in the DG was calculated using the estimated value for basal metabolic rate (BMR) according to the Harris–Benedict equation [15], which was multiplied by a factor of 2.0 according to vigorous active lifestyle as stated by the FAO/WHO/UNU expert consensus document [16]. Once the energy expenditure was estimated, the composition and the adequate percentage distribution of the various macronutrients were calculated for the daily total energy intake, which varied from 2000–2300 kcal in women and 3000–3500 kcal in men. The diet was isocaloric. Carbohydrates were taken as a standard reference value and represented 50% of the total calories in the meal plan. Proteins, on the other hand, were calculated based on the number of training sessions per week and the training load that subjects followed during the CrossFit sessions. Protein sources were mainly fish, legumes, and white and red meats. The latter consumed only twice a week. Protein demand ranged from 1.4 g x kg to up to 2 g x kg of weight. The remaining percentage of calories was calculated in fats, which were mostly mono- and polyunsaturated fats in the form of extra virgin olive oil, nuts, seeds, and omega-3 from fish.

The diet for the subjects in the study was personalized. The differences were in the portions of the food given, in kcals of each subject, but the same foods were used for all participants in the DG. The foods used were mainly present in the MD, preferring foods with whole carbohydrates and a low glycemic index. In this study, it was chosen not to allow supplements of any kind (proteins, amino acids, creatine, etc.). All subjects involved also reported that they had not taken dietary supplements in the past.

The study took place in a gym equipped specifically for CrossFit training, in which all subjects continued to follow their regular CrossFit training program without any variation. Each training session was structured as a classic workout with a specific warm-up phase (10 min), a 1-h central phase, and a cool-down phase (10 min). During the warm-up specific exercises to prepare the body for the central phase were performed. The central phase was structured into two parts: (1) Strength Training of a specific exercise (e.g., front squat, push press, pull-up, bench press, snatch, and deadlift); and (2) Workout of the Day (WOD): a circuit training composed of a specific succession of CrossFit exercises to target metabolic response (e.g., burpees, box jump, jump squat, thruster, push-up, pull-up, and chin-up). The cool-down was administered at the end of the central phase, in which static stretching exercises were performed. The CrossFit training was carried out under the supervision of a qualified CrossFit coach.

### 2.3. Study Procedures

The study procedures included a preliminary analysis, in which a general medical history, nutritional habits, and information on physical activity levels were collected. After having shown the intervention protocol and obtained informed consent, anthropometric (height, weight, and body mass index (BMI)) values were collected, an assessment of the body composition was carried out through a bioimpedance analysis, and explosive strength, muscular endurance, and maximum anaerobic power of the participants were assessed. The following tests were administered before and after the 8 weeks of intervention to evaluate the effects of the personalized MD on CrossFit athletes’ performance. Tests were carried out in the laboratories of the Sports and Exercise Sciences Research Unit of the University of Palermo, under the supervision of qualified personnel. The Fran, the push-up, and chin-up test to exhaustion were administered at the specialized CrossFit gym under the supervision of a qualified CrossFit coach.

### 2.4. Body Composition Analysis

Through a Body Impedance Analyzer (InBody 320, Biospace, Seoul, Korea), a multi-segmental and multi-frequency measurement was carried out to evaluate the body composition of the participants. Participants were advised to reach our labs after a fasting night or without consuming any drinks or food for at least four hours. Before carrying out the test, participants’ heights were measured with a wall-mounted stadiometer, and were asked to remove socks and any kind of metal jewelry (earrings, bracelets, necklaces, etc.) to avoid interference during the evaluation and allow full conduction through the electrodes. The examination was carried out in an upright position. Participants were advised to maintain the upright position with feet placed on the electrodes and hands fixed on the handle electrodes with arms slightly abducted to avoid contact with the torso. The manufacturers’ recommended predictive equations were used to obtain the BIA outcomes taken into consideration in our study: fat mass (FM), total body water (TBW) (including intracellular (ICW) and extracellular water (ECW)) and dry lean mass (Kg).

### 2.5. Circumferences

Body circumferences are indicators of the transverse dimensions of the body segments. Measurement of the body circumferences was evaluated in the following segments: waist, hips, arms, and legs (both left and right upper and lower extremities) were evaluated by following the guidelines proposed by Gibson [17]. The measurements were carried out pre- and post-intervention by the same expert to reduce evaluation biases.

### 2.6. Wingate 30-s Test

The Wingate 30-s test is considered the gold standard to evaluate maximum anaerobic power [18]. The test was performed using a Monark 828E cycle ergometer (Monark excercise AB, Vansbro, Sweden) with computerized data acquisition. The seat and handlebars were adjusted according to the participants’ leg height. Each participant started to pedal at zero resistance to become familiar with the bike and then a two-minute warm-up was administered. During the warm-up phase, the assessor adjusted the resistance on the ergometer to 5% of the participant’s body weight. At the end of the warm-up, the test started and the participant began to ride as fast as he/she could in order to produce maximum speed and power. Immediately, via software, the resistance dropped, increasing the effort needed to sustain the exercise. Participants were advised to maintain speed and power during the 30 s to avoid interruption. At the end of the 30 s, the recording was interrupted automatically and the resistance was removed in order to allow slow pedaling as a cool-down phase. A 2–5-min cool-down was administered according to the participant’s recovery.

During this test, peak power and its derived measures (PP/kg), time at PP (tPP), maximal speed (Vmax), power at Vmax (*p*-Vmax), time at V-max (t V-max), average power and its derived measures (AP/kg), minimum power (MP), power drop (PD) and its derived measures (PD/kg, PD/s, PD/s/kg, and %), as well as power decline (calculated as PP–power at the end of the test), were produced.

### 2.7. Optojump

After a rest of at least 48 h, a test battery was carried out with the Optojump device (Microgate, Mahopac, NY, USA). Optojump is an optical detection system consisting of two bars: a transmitter and a receiver. Each bar contains 96 LED sensors that continuously communicate with each other; they allow the calculation of the duration, height, flight, and contact times during jumps through the software, Optojump Next. The tests performed were: squat jump test, countermovement jump test (performed twice: with free hands and with hands-on waist) [19], and 30-s jump test [20]. Each test was performed 3 times and the highest value was retained for the analysis, excluding the 30-s jump test, which was performed one time at the end of the battery. A proper warm-up was provided between trials and tests to allow for complete recovery and to avoid muscular fatigue.

### 2.8. Squat Jump Test

The squat jump (SJ) allows you to evaluate the explosive strength of the extensor muscles of the lower limbs. It consists of performing a vertical jump at maximum intensity starting from the half squat position (angle of about 90°) without countermovement [19].

### 2.9. Countermovement Jump Test

The countermovement jump test (CMJ) is a test used to evaluate the explosive-elastic strength of the extensor muscles of the lower limbs. The difference with the squat jump is the presence of the countermovement, which allows the recruitment of the elastic force generated. The test was carried out in two variants: with the hands locked to the hips and with the arms free [19].

### 2.10. 30-s Jump Test

The test consists of jumping repeatedly at maximum possible power for 30 s. It evaluates the maximum anaerobic power. The test is carried out with the same methodology as the CMJ, with free arms [20].

### 2.11. Specific CrossFit Performance

Chin-up and push-up tests were performed to evaluate the muscular endurance of the participants. These two tests consisted of performing the maximum number of repetitions possible. Additionally, a Fran workout [21] was used as a test, and the time to complete all the prescribed sets was retained as the dependent variable.

### 2.12. Push-Up Test to Exhaustion

The push-up test to exhaustion was performed on a hard floor, with the hands slightly wider than shoulder-width and the fingers pointing forward with the body horizontal to the surface. One repetition was considered correct when the elbow reached at least 90° flexion during the eccentric phase and a complete extension during the concentric phase. The test ended when participants were no longer able to achieve additional repetitions or when they could no longer maintain the correct techniques.

### 2.13. Chin-Up Test to Exhaustion

The chin-up test to exhaustion was performed on a raised bar with a supine grip and fully-extended elbows with the hands slightly wider than shoulder-width as a starting point. A correct repetition was counted each time the participant was able to raise the body, bringing the chin over the bar during the concentric phase and then returning to the starting position during the eccentric phase of the movement.

### 2.14. “Fran” Training

The “Fran” training consists of 3 rounds of 21-15-9 repetitions of thrusters and pull-ups, respectively [21]. In detail, the participants had to complete 21 repetitions of thrusters and 21 repetitions of pull-ups in the first round, then 15 repetitions for both exercises in the second round, and 9 reps again for both exercises in the third round. The time needed to complete all repetitions was selected as an evaluation parameter. The thruster is defined as the union of the front squat with a push press, performed in a rapid sequence. During the first phase, the front squat is performed with the barbell positioned above the anterior deltoids, in front of the collarbone and above the breastbone. In this phase, from a standing position, keeping the barbell with elbows in a high position, the participant has to complete a knee flexion reaching at least 90° to perform a valid repetition. In the second phase, after the concentric phase of the front squat is performed, with explosive strength, a push phase starts lifting the barbell over the head until a complete elbow extension is completed. After the push-press eccentric phase, the barbell is placed again at the starting position and the exercise starts again. The pull-up test was performed on an elevated bar with a pronated grip. The starting point was at elbows fully extended with the hands slightly wider than shoulder-width apart. One correct repetition was considered only when the body reached the chin over the bar, during the concentric phase, and then a full return to the starting position is completed, during the eccentric phase.

### 2.15. Statistical Analysis

Statistical tests were implemented using JAMOVI (The jamovi project (2021). jamovi (Version 1.6) [Computer Software]. Retrieved from https://www.jamovi.org, assessed on 19 September 2021). Data are reported as mean ± standard deviation (SD). Normality was assessed using the Shapiro–Wilk test. A paired sample *t*-test or a Wilcoxon rank test, according to normality, were used to test the difference within groups before and after the experimental period. Unpaired sample *t*-test or a Mann–Whitney U test, according to normality, were used to test the difference between groups. The alpha for significance was set at *p* < 0.05.

## 3. Results

Table 1 summarizes the baseline characteristics of the two groups.

Table 2 summarizes the body composition and circumferences of the 22 participants (ten in the diet group and twelve in the control group) completing the 8 weeks of intervention and are included in the final analysis. The comparison of the baseline characteristics of the two groups showed no significant differences.

There were no significant differences in participants’ anthropometric values and body composition parameters after the 8 weeks of intervention in both groups. The DG significantly increased upper and lower limb circumferences after the intervention.

Table 3 shows the results regarding the Wingate test. In detail, although we observed an augmented peak power and PP/kg of body weight in both DG and CG (n.s.) after 8 weeks of MD and CrossFit training, max speed was significantly increased only in the DG group (T0: 787.28 ± 106.5, T1: 835.43 ± 128.9; *p* = 0.043). Also, time PP (tPP) was significantly reduced only in the DG group (T0: 2851.22 ± 896.8 ms, T1: 1620.67 ± 768.9 ms; *p* = 0.007). Furthermore, power drop (PD) (T0: 20,527.14 ± 7230.3 W, T1: 22,230.46 ± 7854.5 W; *p* = 0.025) and its derived measures were significantly higher or tended to be significant only in the DG. It seems that CrossFit training, along with MD, improved max participant speed during this test, thereby allowing a slightly higher PP, which was reached earlier (decreasing tPP) and was more difficult to maintain (increasing PD).

Jump performance and the results regarding the specific CrossFit tests are presented in Table 4. The DG showed significant variations after 8 weeks on squat jump (SJ) performance. Jump height increased significantly in the DG (T0-26.96 ± 5.6 cm, T1-29.61 ± 5.6 cm; *p* = 0.035). Moreover, the jump hang-time during SJ showed a significant increase in the DG group (T0-0.47 ± 0.05, T1-0.49 ± 0.06; *p* = 0.033). No significant differences were noted in the other considered jump tests.

Regarding CrossFit performance, the DG group showed significant improvements during the push-up test to exhaustion, the chin-up test to exhaustion, and during the Fran. The push-up number increased from 36.10 ± 15.3 repetitions to 38.90 ± 15.5 repetitions at the end of the 8 weeks (*p* = 0.001). However, the control group also showed a significant improvement during this test (T0-31.25 ± 15.4, T1-34.50 ± 17.2; *p* = 0.034). Regarding the chin-up test to exhaustion, only the DG showed a significant improvement (T0-11.70 ± 5.6, T1-13.60 ± 6.2; *p* = 0.008). Similar results were noted when considering the Fran test. The time to complete the test at T1 (365.20 ± 166.7 s) was shorter when compared to T0 (476.30 ± 330.1 s) for the DG (*p* = 0.002).

## 4. Discussion

The aim of our study was to test the effects of a MD on CrossFit athletes’ performance and body composition. We did not find any variation in body composition in CrossFit athletes after 8 weeks of MD. Notwithstanding, in the DG (but not in the control group), we observed improvement in circumference measures (Table 2) and significant increases in squat jump performance, power, muscular endurance, and anaerobic power (Table 3 and Table 4). Furthermore, CrossFit specific performance was improved after 8 weeks of MD, allowing participants to target abilities useful to practice this discipline (Table 4).

CrossFit training can be high- to very-high-intensity activity and it can require a massive amount of energy expenditure during training and competitions. Observational studies have demonstrated that CrossFit athletes’ dietary intake is frequently below recommendations [13]. Maintaining a deficit in energy intake could be a valuable strategy to target weight loss reduction in overweight or obese individuals [1]. On the other hand, switching energetic metabolism (from carbohydrate to fat recruitment) could be beneficial for athletes. Thus, different nutritional approaches, including ketogenic/low-carbohydrate approaches, have been investigated in CrossFit populations [22,23,24]. Durkalec-Michalski et al. [22] have shown that, although CrossFit training is feasible during ketogenic diet administration, it did not improve general CrossFit performance. On the other hand, the adoption of a ketogenic diet with a low intake of carbohydrates in a sports discipline in which high intensity and anaerobic metabolism are reached can lead to metabolic imbalances for the athlete [25]. Given this, the dietary approach followed by an athlete can influence the physical performance and, above all, that not all dietary regimens (even if well conducted) are able to guarantee real benefits to the athlete. The case of the ketogenic diet is particularly paradigmatic in this regard [25]. Athletes, depending on the discipline, may need a more sustained energy intake in order to maintain functional muscle tissue (improving performance) and successfully complete training sessions and/or competitions. An individualized dietary approach based on MD, such as the one proposed in our study, could be a valuable strategy in this direction, both to preserve a healthy lifestyle and to grant good performance.

Traditional nutritional approaches are usually combined with supplementation, especially in highly-trained athletes. Those strategies are supported by the hypothesis that athletes may need additional help to sustain the energy intake/expenditure balance, improve performance levels, and mitigate muscular fatigue and improve recovery. This is one of the key points that needs clarification in the future to avoid false beliefs that could pose unknown risks to an athlete’s health; only few studies to date have addressed this issue in CrossFit practitioners [26,27,28,29,30,31]. Caffeine supplementation (before a performance test) seems to not be efficient in performance enhancement (when compared to placebo) [26]. Also, betanine supplementation showed no significant effects after 6 weeks of CrossFit training, as shown by Moro et al. [31]. Fernández-Lázaro et al. evaluated the effects of Tribulus Terrestris L. and placebo supplementation, before and after 6 weeks, on 30 CrossFit male athletes [30]. After the intervention, only the weight lifted during the bench press test was significantly increased in the Tribulus Terrestris L. group compared to placebo [30]. Sadowska-Krępa et al. [28] tested the effects of a 6-week green tea extract supplementation on CrossFit trained males, showing a significantly increased antioxidant capacity, while limited effects were noted in aerobic capacity and brain-derived neurotrophic factor [28]. This study showed how simple supplements, such as green tea extract, might have a role in regulating athletes’ metabolic responses but not athletes’ performance. Similar to our study, a Wingate test was applied by Kramer et al. before and after 6 days of nitrate supplementation on 12 male CrossFit athletes [29]. A significant improvement was noted in peak power, while no differences were noted on strength or specific CrossFit performance tests [29]. These results are in contrast with ours, which showed a not significant increase in PP in both groups. We found a significant increase in max speed and PD and reduced tPP during the Wingate test in the DG. Thus, participants on the MD were able to reach higher PP (ns) in a shorter amount of time (tPP), expressing overall higher speeds and PD. These results could support the hypothesis regarding an augmented overall anaerobic power. However, Kramer et al. did not specify the training parameters or exercises, making it difficult to compare the results with ours [29]. Also, the shorter intervention period proposed by Kramer et al. could be a relevant limitation to detect specific CrossFit performance improvements [29]. In contrast to our results regarding CrossFit performance, the authors, in addition, did not find significant results after the specific CrossFit test (Grace) [28]. These results could be justified considering that in our study, we used a different CrossFit WOD to test participants’ performance. Additionally, the improvements in our Fran test shown by the DG could be consistent with the significantly higher max speed reached during the Wingate test, which was probably derived from improved anaerobic system efficiency. Faster repetitions for each administered exercise during the Fran test could have been crucial to reduce the overall duration of this performance. Again, the shorter intervention period proposed by Kramer et al. could be a relevant limitation to detect specific CrossFit performance improvements [29].

Two recent systematic reviews confirmed that due to low-quality papers available, well-designed investigations are necessary to move further in this area [6,32]. In detail, the reviews from dos Santos Quaresma et al. showed that the only compound that helps performance improvement seems to be sodium bicarbonate [6]. The review from de Souza et al. concluded that insufficient results and high-quality studies are available to draw specific guidelines for CrossFit athletes and practitioner populations regarding both nutritional approaches and supplement administration [32]. We did not allow any supplementation in our study, nor did the enrolled subjects report ever having used dietary supplementation in the past. Our working hypothesis is that the beneficial effects of MD, both for the proportion of macronutrients and for the large availability of micronutrients and natural bioactive compounds, can be the best combination between physical health, muscle health, and performance. The data we have collected seem to point in this direction. MD seems to be adequate to sustain CrossFit athletes’ carbohydrate intake to maintain/improve performance, while avoiding macronutrient supplementations. The MD could, in this sense, be a “diet-integrator”: it is naturally rich in antioxidants and polyunsaturated fats, improving endothelium-mediated vasodilation, muscle perfusion, and, ultimately, helping to optimize motor performance [33].

The individualized dietary approach is another aspect of our study deserving attention: a diet could be more beneficial when macronutrients are accurately balanced to energy intake/expenditure and when subjects’ preferences are also taken under consideration (increasing adherence). The subjects belonging to the CG continued their usual diet during the 8-week intervention, which was was not controlled and not necessarily adequate to meet the needs of an athlete practicing regular high-intensity training; the subjects in the DG, on the other hand, carried out a diet: (1) individualized with regard to daily energy intake and macronutrients needs, (2) with a regular and controlled carbohydrate intake (50% of total daily calories), (3) with a regular and controlled protein intake (1.4–2 g/kg of body weight), and (4) which required all macro- and micronutrients to be derived from the intake of foods typical of the Mediterranean diet. Other authors tested different nutritional approaches. Escobar et al. [34] tested the effects of a moderate (6–8 g/kg/day) vs. low (<6 g/kg/day) carbohydrate intakes in 18 CrossFit athletes for 9 days [34]. Although both groups significantly increased the total number of repetitions post-test during a specific AMRAP test (as many repetitions as possible on 12 min of 30″ box jumps, 6 thrusts with additional weight, and 6 burpees), only the moderate carbohydrate group showed a higher percentage increase [34]. Durkalec-Michalski et al. [35] did not find relevant differences in performance between a vegan diet group and a mixed-traditional diet group after 4 weeks of training in CrossFit participants. The authors demonstrated that a diet individualized to subjects’ characteristics could be beneficial, independent of macronutrient sources [35]. Our results do not appear to go in the same direction, indicating that MD seems to provide performance improvements in CrossFit athletes. Again, the limited sample size and the intervention duration could have been relevant in detecting the outcome differences.

We had a satisfactory study adherence: 22 out of 30 athletes (73.3%) completed the study. Participants that dropped out from the study (6 women, 2 men, mean age 33.5 ± 11.7 years; mean weight 70.0 ± 11.2 kg; mean height 172 ± 9.36 cm; mean BMI 23.5 ± 2.87) were not able to complete the study for personal reasons not related to the study protocol nor related to unwillingness to follow the prescribed dietary indications. Participants that did not complete the study had similar characteristics to those included in the analysis (no significant difference in baseline values).

In our opinion, the strengths of our study are the use of MD, a healthy dietetic approach that has been shown to ensure certain benefits in many areas [7,8,9,10,11,12,34]; the absence of any type of supplementation; the individualized approach; the completion of the study within a single season (summer in this case), given the different availability of foods in the MD depending on the time of year; and the large battery of standardized tests, both general and specific for the CrossFit discipline, that we have evaluated. However, there are some limitations. First of all, the open-label design of the study, which may imply a selection bias between groups: the participants in the DG were those who agreed to receive the administration of the MD, while CG were athletes who preferred to maintain their diet habits, thereby avoiding modification. Adherence to the dietary prescriptions provided is difficult to ascertain, and individual motivation is a key issue—being personally motivated to make lifestyle changes is a key element in achieving adherence to the prescriptions provided. Other studies similar to ours adopted an open-label design [24,35]. Both groups showed no significant difference at baseline, nor did the fat-free mass change during the study, as shown in Table 1; the differences in physical performance observed at the end of the study may be attributed only to the different dietary approaches. Second, the limited sample size and the decision to include both CrossFit practitioners and athletes could have influenced our results, thereby hiding potential performance gains promoted by diet. Third, our results are not currently supported by assessments of biochemical and physiological parameters before and after the intervention that could support the observed changes in physical performance. This deficiency limits the strength of the message that emerges from our findings. Despite the abovementioned limitations, our results have highlighted that the MD approach is safe and efficient in the CrossFit population.

In conclusion, our study suggests that a healthy balanced dietary approach such as the Mediterranean diet could be useful to help CrossFit athletes maintain body composition and promote both general and specific CrossFit performance enhancements (anaerobic power, muscular endurance, and explosive strength during squat jump test). Our results seem to indicate that a dietary approach such as the MD, which has demonstrated relevant health benefits, can be effectively offered to CrossFit athletes/practitioners, resulting in the maintenance (if not improvement) of physical performance and avoiding the recourse of unbalanced diets or supplements with unpredictable long-term health effects. As there is no study to date that has directly compared different dietary approaches (e.g., Mediterranean diet vs. ketogenic diet or others), and in consideration of the limited sample size and the short duration of many available studies, properly designed future studies are required to clarify the remaining uncertainties.

## Figures and Tables

**Table 1 jpm-12-01238-t001:** Baseline Characteristics.

Variable	Diet Group (n:10)	Control Group (n:12)	*p*
Age (years)	38.3 ± 8.9	35.6 ± 8.4	0.283
Weight (kg)	70.7 ± 10.4	72.9 ± 13.8	0.701
Height (cm)	170.7 ± 9.0	173.6 ± 9.8	0.500
BMI (kg*m^2^)	24.2 ± 2.5	24.0 ± 2.5	0.818

Data are presented as mean value ± SD. BMI: “Body Mass Index”.

**Table 2 jpm-12-01238-t002:** Body composition and circumferences in the diet and control group before and after the intervention.

Variable	Diet Group (n:10)			Control Group (n:12)		
	T0	T1	*p*	T0	T1	*p*
Weight (kg)	70.7 ± 10.4	69.3 ± 10.1	0.055	72.9 ± 13.8	72.9 ± 14.5	0.984
BMI (kg/m^2^)	24.2 ± 2.5	23.7 ± 2.4	0.056	24.0 ± 2.5	23.9 ± 2.8	0.954
FM (kg)	9.9 ± 4.8	8.9 ± 4.4	0.084	11.7 ± 3.2	11.4 ± 3.3	0.412
FM (%)	14.1 ± 6.5	13.7 ± 8.7	0.432	16.3 ± 5.0	15.8 ± 4.4	0.260
Dry Lean Mass (kg)	15.9 ± 2.8	15.0 ± 2.5	0.647	16.3 ± 3.6	16.3 ± 3.6	0.933
ICW	27.6 ± 4.9	27.8 ± 4.5	0.556	28.0 ± 6.3	28.1 ± 6.3	0.594
ECW	16.7 ± 2.7	16.7 ± 2.5	0.847	16.9 ± 3.5	17.1 ± 3.6	0.405
Total Water	44.3 ± 7.6	44.5 ± 6.0	0.662	44.9 ± 9.8	45.2 ± 9.9	0.514
BMR (Kcal/day)	1671 ± 224	1676 ± 205	0.638	1692. ± 289	1698 ± 291	0.492
Waist (cm)	77.9 ± 5.0	78.9 ± 5.4	0.208	81.6 ± 7.5	82.1 ± 7.7	0.376
Hip (cm)	97.7 ± 5.3	97.0 ± 5.8	0.373	99.7 ± 6.3	99.3 ± 6.9	0.569
Right Arm (cm)	28.9 ± 2.0	29.8 ± 2.0	0.017	29.4 ± 2.9	29.7 ± 4.2	0.506
Left Arm (cm)	28.8 ± 2.8	29.8 ± 2.9	0.008	29.1 ± 2.8	29.3 ± 3.7	0.700
Right Leg (cm)	53.0 ± 2.9	54.9 ± 3.5	0.019	55.7 ± 4.5	54.9 ± 4.3	0.060
Left Leg (cm)	52.3 ± 2.9	55.5 ± 2.0	<0.001	54.9 ± 4.6	55.3 ± 4.6	0.377

Data are presented as mean value ± SD. BMI: “Body Mass Index”; FM: “Fat mass”; ICW: “Intracellular Water”; ECW: “Extracellular Water”; BMR: predicted “Basal Metabolic Rate”; T0: baseline; T1: end of the protocol.

**Table 3 jpm-12-01238-t003:** Results of the Wingate test.

Variable	Diet Group			Control Group		
	T0	T1	*p*	T0	T1	*p*
Peak Power (W)	14,823.94 ± 3854.47	16,772.78 ± 5830.09	0.067	17,679.70 ± 8785.39	18,830.21 ± 8512.35	0.570
PP/kg (W/kg)	212.41 ± 44.40	237.38 ± 60.41	0.075	233.44 ± 82.87	250.79 ± 79.50	0.403
tPP (ms)	2851.22 ± 896.77	1620.67 ± 768.87	0.007	3143.83 ± 1898.69	1955.17 ± 1080.85	0.129
Vmax (rpm)	787.28 ± 106.47	835.43 ± 128.92	0.043	840.71 ± 172.01	857.87 ± 162.21	0.742
*p* Vmax (W)	5842.77 ± 2205.49	6070.75 ± 2795.01	0.713	6043.11 ± 162.21	6718.03 ± 162.21	0.381
t Vmax (ms)	8770.20 ± 1993.07	6693.40 ± 2739.53	0.084	7514.58 ± 2308.47	6537.67 ± 1967.69	0.204
Average Power (W)	2580.48 ± 505.44	2406.21 ± 613.93	0.076	2619.73 ± 842.97	2615.23 ± 876.24	0.986
AP/kg (W/kg)	37.00 ± 2.93	33.48 ± 6.03	0.110	35.64 ± 6.00	35.98 ± 6.63	0.894
Minimum Power (W)	−4326.83 ± 1782.48	−5450.01 ± 2443.47	0.203	−6225.48 ± 2922.29	−5482.59 ± 2073.96	0.443
MP/kg (W/kg)	−59.93 ± 19.70	−76.68 ± 26.77	0.203	−83.54 ± 35.47	−75.09 ± 22.44	0.434
Power Drop (W)	20,527.14 ± 7230.26	22,230.46 ± 7854.46	0.025	23,582.70 ± 11,445.39	24,312.80 ± 10,441.08	0.762
PD (W/kg)	290.85 ± 76.49	314.35 ± 80.70	0.051	313.43 ± 110.10	325.88 ± 98.41	0.633
PD (W/s)	707.83 ± 249.32	766.57 ± 270.84	0.025	813.20 ± 394.67	838.37 ± 360.04	0.762
PD (W/s/kg)	10.03 ± 2.64	10.84 ± 2.78	0.051	10.81 ± 3.80	11.24 ± 3.39	0.633
PD (%)	136.31 ± 19.23	132.42 ± 7.07	0.846	134.71 ± 12.27	130.58 ± 5.92	0.791
Power decline * (W)	14,108.69 ± 4805.27	18,007.02 ± 7401.43	0.062	17,489.19 ± 9294.05	19,495.74 ± 9344.67	0.395

Data are presented as mean value ± SD. CG: “Control Group”; DG: “Diet Group”; PP: “Peak Power”; Vmax: “Max Speed”; *p*: “Power”; t: “time”; s: “seconds”; AP: “Average Power”; MP: “Minimum Power”; PD: “Power Drop”; W: “Watt”; T0: baseline; T1: end of the protocol. * Power decline = peak power–power at the end of the test.

**Table 4 jpm-12-01238-t004:** Results of jump tests and CrossFit performance.

Variable	Diet Group			Control Group		
	T0	T1	*p*	T0	T1	*p*
SJ (s)	0.47 ± 0.05	0.49 ± 0.06	0.033	0.49 ± 0.08	0.50 ± 0.07	0.142
SJ (cm)	26.96 ± 5.59	29.61 ± 5.59	0.035	30.32 ± 8.63	30.98 ± 7.95	0.169
CMJ (s)	0.49 ± 0.06	0.50 ± 0.06	0.126	0.52 ± 0.08	0.52 ± 0.08	0.504
CMJ (cm)	29.85 ± 7.34	29.85 ± 7.70	0.123	33.36 ± 9.77	33.71 ± 9.08	0.643
CMJ free arms (s)	0.52 ± 0.07	0.54 ± 0.07	0.111	0.55 ± 0.08	0.55 ± 0.08	0.398
CMJ free arms (cm)	34.17 ± 8.36	35.84 ± 8.36	0.121	37.75 ± 10.02	38.13 ± 9.77	0.618
30 s SJ test (n)	24.30 ± 2.91	23.70 ± 2.67	0.347	23.42 ± 2.31	23.67 ± 1.97	0.515
M 30 s SJ test (cm)	25.74 ± 6.95	26.21 ± 6.83	0.450	27.67 ± 8.64	27.35 ± 7.36	0.718
M 30 s SJ test (W)	17.12 ± 3.57	17.11 ± 3.27	0.990	17.93 ± 4.57	17.84 ± 3.93	0.814
30 s max high (cm)	31.96 ± 3.27	32.31 ± 7.18	0.711	35.12 ± 9.80	34.90 ± 8.69	0.797
30 s max power (W)	20.34 ± 4.21	20.41 ± 3.74	0.904	21.95 ± 5.51	21.95 ± 4.340	0.624
Fran (s)	476.30 ± 330.14	365.20 ± 166.70	0.002	455.13 ± 107.50	451.00 ± 97.47	0.911
Push-up test (n)	36.10 ± 15.28	38.90 ± 15.47	0.001	31.25 ± 15.43	34.50 ± 17.23	0.034
Chin-up test (n)	11.70 ± 5.60	13.60 ± 6.24	0.008	8.88 ± 6.51	9.63 ± 6.32	0.080

Data are presented as mean value ± SD. SJ: “Squat Jump”; CMJ: “Counter Movement Squat Jump test”; M: “Mean”; W: “Watt”; s: “seconds”; cm: “centimeters”; T0: baseline; T1: end of the protocol.

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
