# Peer review of "Effects of Mediterranean Diet Combined with CrossFit Training on Trained Adults’ Performance and Body Composition"

_jpm, 2022, doi:10.3390/jpm12081238_

Round 1

Reviewer 1 Report

This is an interesting study on the usefulness of the Mediterranean diet on the performance and body composition of athletes who practice Crossfit.

However, there are 3 major methodological considerations that would lead to rejecting the publication:

1- The choice of the sample. It is a convenience sample, based on the subjects' willingness to change their diet, which implies an important bias.

2- No data is shown about the athletes who after starting the study did not finish it and if they were different from those who finished it.

3- The explanation about the calculation of the energy expenditure of athletes is insufficient and incomplete. There are methods to estimate it easily.

Author Response

REVIEWER #1

We would like to thank you for the detailed review of our manuscript. We greatly appreciate the effort you made concerning your critique for the review of our study. We are aware of the relevance of the methodological criticisms raised, and we strongly hope that the clarifications and the additions we have made in support of our approach can be deemed valuable.

This is an interesting study on the usefulness of the Mediterranean diet on the performance and body composition of athletes who practice Crossfit.

However, there are 3 major methodological considerations that would lead to rejecting the publication:

1- The choice of the sample. It is a convenience sample, based on the subjects' willingness to change their diet, which implies an important bias.

We definitively agree with the reviewer that this is a critical point in our study. In support of our decision to divide the sample in relation to individual willingness to change their dietary habits and not randomize it, we would like the reviewer to consider the following:

- Many data concur in indicating that adherence to lifestyle modification is a key element in the success of the intervention. In our study, which enrolled healthy young people, individual compliance and willingness to commit to adhere to the instructions to modify their dietary habits was critical.

- Other studies carried out in settings similar to ours have opted for a open label approach thus not randomizing the sample (Kephart, W.C.; et al. The Three-Month Effects of a Ketogenic Diet on Body Composition, Blood Parameters, and Performance Metrics in CrossFit Trainees: A Pilot Study. Sports (Basel) 2018, 6, doi:10.3390/sports6010001.   -     Durkalec-Michalski, K, et al. Effect of a Four-Week Vegan Diet on Performance, Training Efficiency and Blood Biochemical Indices in CrossFit-Trained Participants. Nutrients 2022, 14, 894. doi: 10.3390/nu14040894).

- Our analysis shows, however, that the decision (voluntary or not) to adopt a controlled, Mediterranean-type diet in CrossFit athletes is likely to produce benefits, moving athletes away from do-it-yourself diets or supplementation potentially unhealthy if not dangerous. In our view, this is a message of no minor relevance.

The open-label design had already been stated as a possible limitation of the study; in the revised version we have more thoroughly pointed out the possible limitation related to the study design by arguing more extensively the reasons for our choice

2- No data is shown about the athletes who after starting the study did not finish it and if they were different from those who finished it.

Thank you for the suggestion. We have added a paragraph in the discussion section in which we analyze the issue of protocol adherence; we have included data about subjects who dropped out of the study and the reasons for leaving.

The following table, not added to the text for conciseness, but reporting data that may be available along with any data from the initial assessment of subjects who did not complete the study, supports the above.

Comparison between baseline characteristics of Included and Dropped out participants

Variable

Included

Drop out

p

Age (years)

38.2±8.62

33.5±11.7

0.255

Weight (kg)

71.8±12.0

70.0±11.2

0.716

Height (cm)

172±9.28

172±9.36

0.980

BMI (kg*m-2)

24.1±2.34

23.5±2.87

0.583

FM (kg)

10.8±4.08

12.0±4.74

0.505

FM (%)

15.2±5.74

16.9±5.29

0.495

Dry Lean Mass (kg)

16.1±3.12

15.5±2.47

0.684

Metabolic Rate

1682±252

1624±196

0.563

3- The explanation about the calculation of the energy expenditure of athletes is insufficient and incomplete. There are methods to estimate it easily.

The reviewer is right. We have now presented more in detail the procedure for calculation of energy expenditure that is based on FAO/WHO/UNU factor for very active/sportive people of 2 multiplying the Harris Benedict predicted BMR.

We hope that we have successfully changed our manuscript according to your suggestions and that we have provided all the necessary explanations. We also hope that the manuscript now fulfills your criteria, and the Journal criteria for publication.

Reviewer 2 Report

Interesting subject with a high level of novelty.

The method is described sufficiently.

Some additional description of the diet plans drafted for the diet group, would be important.

Statistical analysis is conducted in the appropriate manner and results presented well.

Some minor language check, however is required.

Author Response

REVIEWER #2:

We would like to thank you for your expert review of our manuscript. Thank you very much for your positive opinion regarding our manuscript. We put a lot effort in this study and we appreciate your opinion very much.

Interesting subject with a high level of novelty.

The method is described sufficiently.

Thank you for your comment. We have further revised the methods section to improve the overall quality of our manuscript.

Some additional description of the diet plans drafted for the diet group, would be important.

Thanks for the suggestions. We have described more in detail the dietary program based on the Mediterranean Diet that we proposed to the Diet Group and additional information has been added as requested.

Statistical analysis is conducted in the appropriate manner and results presented well.

Some minor language check, however is required.

Dear Reviewer, the entire manuscript has been checked and typing/language errors have been corrected, thanks.

We hope that we have successfully changed our manuscript according to your suggestions and that we have provided all the necessary explanations. We also hope that the manuscript now fulfills your criteria, and the Journal criteria for publication.

Reviewer 3 Report

The manuscript under evaluation reports the effect of Mediterranean diets on crossfit athletes. There are certain issues that I need clarification from the authors and a revision of the manuscript before I can proceed with proper evaluation. The authors may choose to provide a rebuttal to my comments and/or furnish their manuscript with the pertinent information.

1) It is unclear through out the manuscript the major difference between the control diet and MD. A thorough information (%source of energy) regarding formulation of the diets (Control and MD) must be provided.

2) The goal/necessity to an MD approach for crossfit athletes is also not clear from the introduction. Please add sufficient information regarding past studies if available - why MD is beneficial over traditional personalized dietary supplements that enhance performance of professional athletes practicing crossfit.

3) Lines 106-116: I am finding it difficult to comprehend this section. Is there any better way to represent this? Kindly check. If this is pertinent to the relevant diet information, please find a better way to present.

4) Is it possible to provide the anthropometric characteristics of the participants in a table form? Also, an exercise key (figure) representing the different tests performed would be very useful.

5) Table 2: Only 4 out of 13 parameters tested result in significant difference for MD. Is this significant enough to make a conclusive statement about anaerobic power? Besides, are these parameters tested independent of each other? Similar question arises for Table 3 as well.

6) Discussion line 329 - kindly point out the exact tables/data supporting this sentence.

7) A few major limitations must be addressed for this study - I  recommend the authors to provide any blood analysis/physiologic analysis (if available) to be supplemented with this manuscript. Also, the size of the groups, history of athletic performance, consumption of supplements prior to the study that may affect the study period etc must be mentioned for normalization wrt to the present study. As a matter of fact, in the absence of any physiologic data, it makes the conclusions of the manuscripts very much open to interpretations. Kindly enhance the discussion/conclusions in a way these raised issues may be addressed.

Author Response

REVIEWER #3

We would like to thank you for the detailed review of our manuscript. We greatly appreciate the effort you made concerning your critique for the review of our study. We have accepted all your suggestions and revised the article according to them.

The manuscript under evaluation reports the effect of Mediterranean diets on crossfit athletes. There are certain issues that I need clarification from the authors and a revision of the manuscript before I can proceed with proper evaluation. The authors may choose to provide a rebuttal to my comments and/or furnish their manuscript with the pertinent information.

1) It is unclear through out the manuscript the major difference between the control diet and MD. A thorough information (%source of energy) regarding formulation of the diets (Control and MD) must be provided.

Thank you for the helpful suggestion. We have revised the manuscript both in the materials and methods section and in the discussion. The intervention section explains in detail the criteria used and the characteristics of the dietary plan to which the subjects in the "Diet Group" were assigned. As for the control group, which were not given specific instructions but continued their usual diet, this was variable from day to day, unbalanced and of irregular caloric intake making it impossible to compare with a balanced diet both in terms of % of macro- and micronutrients and daily caloric intake such as the individualized one that was proposed to the "Diet Group."

2) The goal/necessity to an MD approach for crossfit athletes is also not clear from the introduction. Please add sufficient information regarding past studies if available - why MD is beneficial over traditional personalized dietary supplements that enhance performance of professional athletes practicing crossfit.

Thank you for raising this point that allows us to further clarify that there are available only a few low-quality studies that have addressed the effects of diet and/or dietary supplementation in CrossFit athletes. We modified the introduction according to the reviewer's suggestions. We have also added a final conclusive paragraph “As there is no study to date that has directly compared different dietary approaches (e.g., Mediterranean diet vs. ketogenic diet or other), it is certainly an interesting cue that deserves attention in the future”.

3) Lines 106-116: I am finding it difficult to comprehend this section. Is there any better way to represent this? Kindly check. If this is pertinent to the relevant diet information, please find a better way to present.

We agree with the reviewer. The paragraph has been entirely rewritten, and the technical and theoretical basis of the dietary therapy intervention and the operational methods by which it was implemented in every individual enrolled we hope that now may be clearer.

4) Is it possible to provide the anthropometric characteristics of the participants in a table form? Also, an exercise key (figure) representing the different tests performed would be very useful.

Dear Reviewer, the anthropometric characteristics of the participants have been summarized as suggested in a new added Table 1.

The tests that were conducted are highly standardized and the methodology of execution is known to those in the field, but we agree with the reviewer regarding the desirability of making it easier for any reader to understand the tests with illustrative pictures. Figures representing our performance tests have been provided as suggested.

5) Table 2: Only 4 out of 13 parameters tested result in significant difference for MD. Is this significant enough to make a conclusive statement about anaerobic power? Besides, are these parameters tested independent of each other? Similar question arises for Table 3 as well.

Dear Reviewer, regarding the Wingate test, although we did not find a significant improvement in Peak Power (and derived PP/kg) in MD, we supported the hypothesis regarding an increased anaerobic power due to improvement in tPP (time to reach PP) that was reduced along with increased participants speed and power drop. This implies that participants reached PP faster (lower tPP and higher speed) and dropped more power during the test (higher Power drop, mainly due to increased ns PP). The results recorded during the Wingate test were calculated from the same software during the same test, thus are not independent from each other. This allowed us to support anaerobic power improvements even though only 4 parameters reached significance in MD. Notwithstanding, we recognize that the discussion on these parameters was not clear thus we modified the manuscript accordingly.

Regarding table 3 data (now referred to as table 4), also jump tests were recorded using the same software but not using the same test so parameters are not dependent from each other. Only when considering the jump heigh and time referred to the same test, parameters are dependent (SJ-CMJ-CMJ free arms). Additionally, for the 30-s jump test, 5 different parameters were recorded and calculated (dependent on each other). CrossFit variables were completely independent between each other and between other tests. Thank you.

6) Discussion line 329 - kindly point out the exact tables/data supporting this sentence.

Dear Reviewer, references to tables were added where needed as requested. Thanks..

7) A few major limitations must be addressed for this study - I  recommend the authors to provide any blood analysis/physiologic analysis (if available) to be supplemented with this manuscript. Also, the size of the groups, history of athletic performance, consumption of supplements prior to the study that may affect the study period etc must be mentioned for normalization wrt to the present study. As a matter of fact, in the absence of any physiologic data, it makes the conclusions of the manuscripts very much open to interpretations. Kindly enhance the discussion/conclusions in a way these raised issues may be addressed.

Thank you for the suggestion. We are aware of the limitations of our study, in the revised version we have stressed the need for caution in the interpretation of our results as properly recommended. Information regarding the athletic history, previous dietary habits and the statement that no subject evaluated as using supplements even before the study as requested in the methods and discussion section (lines 78-79, lines 397-399). Unfortunately, we did not collect additional physiological data considering our healthy athletic sample and our aim (effects of MD on performance and body composition). In order to do that we applied the Wingate test, which is commonly used to evaluate anaerobic power in athletes along with other laboratory and field-based standardized tests. We recognize that without further physiological data our conclusions must be thoughtfully interpreted.

The discussion and the conclusions were considerably modified accordingly. We hope that the revised discussion and the improved conclusions addresses this issue properly.

We hope that we have successfully changed our manuscript according to your suggestions and that we have provided all the necessary explanations. We also hope that the manuscript now fulfills your criteria, and the Journal criteria for publication.

Round 2

Reviewer 1 Report

The authors' points are convincing and I think the article could be accepted. It highlights the importance of a healthy diet (in this case a Mediterranean diet recommended by a dietitian) for improving sports practice and body composition.

Author Response

We would like to sincerely thank the reviewer for his/her expert review of our manuscript without which we would not have been able to achieve such a strong result and for his/her fully positive opinion of the revised version.

Reviewer 3 Report

Many thanks to the authors for their careful consideration in revising the manuscript. Overall the article seems to have solid foundation. I have a few recommendations that the authors might consider for further improvements:

1) Absence of physiological data from the participants in support of the bodily changes that improved their Crossfit output has to be addressed as a limitation of the current study. Kindly consider.

2) It is unclear the major changes in diet that was caused by MD in comparison to the control diet or normal diet that the control group consumed. Once again, I request the authors to kindly chart out a critical discussion, if possible the dietary differences that possibly enhance anerobic or explosive performances.

3) There are counter argument articles such as doi.org/10.1186/1550-2783-9-34 available that poses an opposite view of the current findings by the authors. I request strengthening the discussion with such opposing findings, which can obviously be negated by the sample size/study design. 

4) The actual pictures of the crossfit training seems to be a misfit. My recommendation for first round review was based on a broad readership who may not be acquainted with the exercises.  If a schematic representation is not available, kindly remove the pictures.

Author Response

REVIEWER #3

We would like to thank again you for the further evaluation of our manuscript. We greatly appreciate the effort you made concerning your critique for the review of our study. We have accepted all your suggestions and revised the article according to them.

Many thanks to the authors for their careful consideration in revising the manuscript. Overall the article seems to have solid foundation. I have a few recommendations that the authors might consider for further improvements:

1) Absence of physiological data from the participants in support of the bodily changes that improved their Crossfit output has to be addressed as a limitation of the current study. Kindly consider.

Thank you for the suggestion: we have clearly acknowledged this issue as a limitation of the study adding the following sentence in the discussion section: “our results are not currently supported by assessments of biochemical and physiological parameters before and after the intervention that could support the observed change in physical performance. This deficiency limits the strength of the message that emerges from our findings”.

2) It is unclear the major changes in diet that was caused by MD in comparison to the control diet or normal diet that the control group consumed. Once again, I request the authors to kindly chart out a critical discussion, if possible the dietary differences that possibly enhance anerobic or explosive performances.

We want to thank again the reviewer for this appropriate comment. In the revised version, we more schematically described the major differences in the two dietary patterns that our athletes followed, also adding the following in the discussion: “The subjects belonging to the CG continued in the 8-week intervention their usual diet that was not controlled and not necessarily adequate to meet the needs of an athlete practicing regular high-intensity training; the subjects in the DG, on the other hand, carried out a diet: 1) individualized with regard to daily energy intake and macronutrients needs, 2) with a regular and controlled carbohydrate intake (50% of total daily calories), 3) with a regular and controlled protein intake (1. 4-2 g/kg of body weight), 4) which required all macro- and micronutrients to be derived from the intake of foods typical of the Mediterranean diet”.

3) There are counter argument articles such as doi.org/10.1186/1550-2783-9-34 available that poses an opposite view of the current findings by the authors. I request strengthening the discussion with such opposing findings, which can obviously be negated by the sample size/study design.

Thank you for the comment. This issue is of particular relevance. The evidence provided by the authors in the article by Paoli et al, cited by the reviewer, as well as that in Durkalec-Michalski et al, cited in our paper (see ref 24), shows that the dietary approach followed by an athlete can really affect the physical performance and, above all, that not all dietary regimens (even if well conducted) are able to guarantee real benefits to the athlete. The case of the ketogenic diet, which, moreover, our group studied (see ref 25 in the text) is particularly paradigmatic in this regard. In agreement with the reviewer's suggestion, we further emphasized this fundamental aspect by modifying the discussion section.

The limited sample size and the short duration of the available studies is repeatedly cited in the discussion as a possible limitation in the strength of the data reported by many authors.

4) The actual pictures of the crossfit training seems to be a misfit. My recommendation for first round review was based on a broad readership who may not be acquainted with the exercises.  If a schematic representation is not available, kindly remove the pictures.

Thank you for the suggestion. As we also pointed out in the text of the article, the physical tests carried out in our study are all highly standardized and the protocol used to carry them out is well known to practitioners. Nevertheless, to meet the very understandable requests for more clarity for not experts raised from this (but also from another) reviewer, we added the photographs that have now been removed. We think that the understanding of the study procedures remains equally well understood by the reader

We hope that we have successfully changed our manuscript according to your suggestions and that we have provided all the necessary explanations. We also hope that the manuscript now fulfills your criteria, and the Journal criteria for publication.